# Nurses and physicians attitudes towards factors related to hospitalized patient safety

Iwona Malinowska-Lipień[1]*, Agnieszka Micek[1], Teresa Gabryś[1], Maria Kózka[1], Krzysztof Gajda[2], Agnieszka Gniadek[1], Tomasz Brzostek[1], Allison Squires[3]

1 Institute of Nursing and Midwifery, Faculty of Health Sciences, Jagiellonian University–Medical College, Krakow, Poland, 2 Institute of Public Health, Faculty of Health Sciences, Jagiellonian University–Medical College, Krakow, Poland, 3 Rory Meyers College of Nursing, New York University, New York, New York, United States of America

* iwona.malinowska-lipien@uj.edu.pl

**Data Availability Statement:** All relevant data are within the manuscript and its Supporting Information files.

## Abstract

### Introduction

The attitudes of healthcare staff towards patients' safety, including awareness of the risk for adverse events, are significant elements of an organization's safety culture.

### Aim of research

To evaluate nurses and physicians' attitudes towards factors influencing hospitalized patient safety.

### Materials and methods

The research included 606 nurses and 527 physicians employed in surgical and medical wards in 21 Polish hospitals around the country. The Polish adaptation of the Safety Attitudes Questionnaire (SAQ) was used to evaluate the factors influencing attitudes towards patient safety.

### Results

Both nurses and physicians scored highest in stress recognition (SR) (71.6 and 80.86), while they evaluated working conditions (WC) the lowest (45.82 and 52,09). Nurses achieved statistically significantly lower scores compared to physicians in every aspect of the safety attitudes evaluation (p<0.05). The staff working in surgical wards obtained higher scores within stress recognition (SR) compared to the staff working in medical wards (78.12 vs. 73.72; p = 0.001). Overall, positive working conditions and effective teamwork can contribute to improving employees' attitudes towards patient safety.

### Conclusions

The results help identify unit level vulnerabilities associated with staff attitudes toward patient safety. They underscore the importance of management strategies that account for staff coping with occupational stressors to improve patient safety.

**Funding:** Funding was provided by the Faculty of Health Sciences, Jagiellonian University Medical College. Krakow, Poland (No. K/ZDS/007928). The funders had no role in study design, data collection and analysis, decision to publish, or preparation of the manuscript.

**Competing interests:** NO - The authors have declared that no competing interests exist.

## Introduction

Patients' safety constitutes an important aspect of health care delivery. The goal of patient safety programs is to prevent errors and reduce the potential for damage suffered by patients while receiving healthcare services. To sustain patient safety practices by staff, constant training and reinforcement of it for healthcare staff is required to avoid adverse events.

The International Classification for Patient Safety (ICPS) defines 'patient safety' "as the act of avoiding, preventing or improving adverse outcomes or injuries occurred throughout the medical-hospital process" [1]. According to the World Health Organisation (WHO), adverse events related to health care are one of the substantial reasons for death and disability of hospitalised patients. For the WHO, "patient safety means the reduction to an acceptable minimum level of risk of unnecessary harm related to health care".

Medical errors affect health care systems around the world. The cost related to medical errors worldwide is estimated at USD 42 billion yearly [2]. Research has shown that the number of fatalities caused by medical errors in the USA every year exceeds 250,000, which makes them the third largest cause of death [3]. In low- and middle-income countries, however, every year in hospitals 134 million adverse events take place and 2.6 million result in death [1]. Analyses of European data—mainly from Denmark, France, Spain [1] found that medical errors and adverse events related to health care occur in 8–12% of hospitalisations. An OECD report concluded that 15% of hospital operating costs may be attributed to the treatment of adverse events [4]. Actions aimed at limiting causes of adverse events occurring during hospitalisation may improve patients' health outcomes and lead to financial savings for healthcare organizations and national health systems [5].

Hughes et al. noted the importance of staff's attitudes when creating a work environment where patient safety is a high priority [6]. Understanding medical staff's attitudes towards factors conditioning patients' safety are crucial to improve patients' care and safety. Analyzing factors that contribute to the occurrence of adverse events helps create the conditions that foster changes in staff behaviours, which may make the healthcare environment safer [7]. Standardized systems with rules and procedures focused on patient safety for both personnel providing direct care and their managers also helps minimize the risk of personnel making mistakes [8]. Further, using tools like the Safety Attitudes Questionnaire (SAQ) for assessing healthcare staff attitudes about patient safety at the organizational level is an important part of fostering organizational change to improve patient safety [9–19]. It is also worth remembering that a patient's safety during treatment is inherently related to the activities of people involved in delivering it—from nurses to physicians to administrators. High awareness of occupational safety, cooperation in an interdisciplinary team, assessment of the culture of work safety, as well as analysis and drawing conclusions may increase the quality and safety in real terms, and make the patient feel safer. On the other hand, it should be remembered that a safety culture is not only important for reducing the risk for harm to hospitalized patients, but is also key to ensuring a safe working environment for healthcare professionals. A first step toward building safety cultures is to conduct research to assess healthcare workers' attitudes towards patient safety.

Poland has nascent infrastructure for monitoring patient safety. First, there is no mandatory reporting requirement for the numbers and types of medical errors. As a result, no systematic monitoring in this area is performed and the true scale of adverse events in hospitals in Poland is not precisely known. Studies using an internationally standardised tool have also not been performed to date. This study is a first step toward generating evidence to identify where organizations in Poland can direct efforts to improve patient safety.

## Aim of research

To determine Polish nurses' and physicians' attitudes about patient safety practices in hospitals.

Detailed objectives:

1. What attitudes towards factors related to hospitalized patient safety are presented by nurses and by doctors?

2. What were the differences in attitudes towards safety in the group of nurses and doctors?

3. To what extent did the type of ward differentiate the attitudes towards safety of nurses and doctors?

4. What was the relationship between the respondents' gender and attitudes towards safety?

5. What was the relationship between the time of employment of the respondents and their attitudes towards safety?

## Materials and methods

This was a descriptive, cross-sectional study carried out on a group of nurses and physicians employed in surgical and medical wards in 21 Polish hospitals, located in different parts of the country. The selection of hospitals was based on a stratified selection procedure that accounted for geographic and service administrative area factors along with population density and hospital reference level, similar to those used in the RN4CAST methodology [20]. The study included only state multi-profile hospitals, serving patients 24h/7day care. The studies were performed in the years 2018–2019 after obtaining the consent of the Bioethical Commission of Jagiellonian University (KBE UJ) No. 1072.6120.111.2018.

### Sample

The inclusion criteria were a) Polish national, b) employed at a study site, and c) actively working during the study as a nurse or physician in a given hospital. All other hospital staff was excluded from the study. Nurses on maternity leave, extended sick leave or study leave were excluded from participation.

The sample size necessary to detect differences in the mean percentage results of the safety attitude subscales between physicians and nurses was calculated, assuming equal numbers of doctors and nurses in the groups, 95% power and the *FWER* (*family-wise error rate*) value at the 0.05 level. It was shown that 1,050 persons would be sufficient to detect a small-size effect (eta squared 0.02), according to Cohen's recommendations [21].

### Instrument

The study was performed with the diagnostic survey method using the survey technique with SAQ-SF PL tool in the Polish adaptation by Malinowska-Lipień et al. [22]. The Safety Attitudes Questionnaire reliability had a Cronbach's Alpha of 0.98. Before performing the analysis validity of the Polish adaptation of SAQ-SF, the Kaiser test was used to check whether the data meet the requirements of the factor analysis. The Kaiser-Mayer-Olkin (KMO) value, being the measure of the adequacy of the sample selection, was estimated at the level of 0.87 (df = 8630, p<0.001). This model explained 68% of the total variance of the analysed set of variables [22].

The instrument consists of 41 entries, divided into two parts, with part two consisting of a demographic profile. The first part contains 36 questions subdivided into six subscales. First,

1/ Teamwork climate—TC (questions from 1 to 6), which evaluates the perception of cooperation quality among staff; 2/Safety climate—SC (questions from 7 to 13)–evaluates the perception of employees' organisational involvement in patient's safety; 3/Job satisfaction—JS (questions from 15 to 19)–evaluates subjective feeling connected to professional experience; 4/ Stress recognition—SR (questions from 20 to 23)–evaluates of the influence of stressors on work efficiency; 5/Perception of management–PM), evaluates at the ward and hospital level (questions from 24 to 28); and finally 6/Work conditions—WC) (questions from 29 to 32), which concern the quality of environmental and logistic support in the workplace (e.g. appliances, equipment and professionals). Five questions in part one are not included in any of the subscales, i.e. question 14 related to the assessment of a managing staffer in the context of providing safety, and questions from 33 to 36 concerning the evaluation of conflicts and cooperation among the members of the interdisciplinary team, i.e. nurses, doctors, pharmacists.

Answers are scored on a 5-point Likert scale (1 = strongly disagree (A); 2 = rather disagree (B); 3 = neutral answer (C); 4 = quite agree (D); 5 = strongly agree (E)), while questions 2, 11 and 36 were scored reversely. For each question, the questionnaire authors included the "does not concern" option. To calculate the score according to the diagnostic key, the conversion to the 100-point was implemented, i.e.: 1 = 0; 2 = 25; 3 = 50; 4 = 75; 5 = 100. The final score of the questionnaire takes the value from 0 to 100 points, where zero means the worst and 100 pts.– the best attitudes towards factors conditioning patients' safety. Scores at the level of 75 pts. and higher are considered as a positive attitude in the area covering a particular subscale [23].

## Data collection

The SAQ-SF was used in the research' to evaluate attitudes of nurses and physicians towards factors patients' safety. The survey questionnaire was independently filled in by nurses and physicians, who voluntarily agreed to participate. All potential participants were informed in writing and verbally by the hospital coordinator about the research aim and anonymity. In each hospital, a study project coordinator was responsible for data collection. In clinical departments, the questionnaire with an envelope attached to it was distributed during departmental staff meetings led by the hospital project coordinator. Participants had 4 weeks to complete the questionnaire. Completed questionnaires packed in a sealed plastic envelope were deposited into a secure box. After 4 weeks, the coordinator was responsible for collecting the boxes, securing them, and handing them over to the research team. Participants were informed that participation was voluntary and anonymous, that all responses would be kept confidential and that no individual responses would be available to hospital management.

## Statistical analysis

All analyses were carried out using the R software version 3.6.1 (Development Core Team, Vienna, Austria). The statistical significance level was set at $\alpha = 0.05$. The descriptive statistics for each subscale of SAQ was presented as average (x) and standard deviation (SD). For each respondent, mean values were independently calculated within each SAQ subscale. In the case respondents pointed to the "does not concern" answer, it was ignored while calculating the mean result of the scale. Questionnaires with over 10% of incomplete answers to the 36 questions of the SAQ were excluded from analyses [23].

**The dependent variable** was defined as the attitudes towards safety measured with the standardised SAQ-SF.

**The independent variables** included: professional group (nurses and physicians), gender, type of ward (surgical or medical unit), age category of the treated patients (adults and/or

children), professional experience (<1 year, 1–4 years, 5–10 years, 11–20 years and > 20 years), number of nursing staff, number of physicians, number of beds in the ward.

Uni- and multi-variate analyses were used to compare mean percentage results of the SAQ subscales between the groups defined by categories of the following factors: professional group (nurses and physicians), gender (men and women), ward (surgical and medical unit), age group of the treated patients (adults, children, and both groups), work experience (<1 year, 1–4 years, 5–10 years, 11–20 years and > 20 years). A one-factor multi-dimensional analysis of variants was performed to estimate the influence of each factor separately on the combined set of dependent variables (safety climate, stress recognition, teamwork climate, working conditions, work satisfaction, perception of management). If the result of the one-way MANOVA was statistically significant, the one-way ANOVA analysis was performed, studying separately each dependent variable in order to identify those dependent variables that significantly contributed to obtaining a statistically significant global effect. Bonferroni correction was used for multiple comparisons, resulting in the criterion of rejecting a null hypothesis at a significance level p<0.008 instead of p <0.05 (six dependent variables). To estimate the relation between the number of beds, the number of nurses and the number of doctors in the ward and each of the SAQ subscales, one-dimensional linear and logistic regression models of multiple variables were used, standardised for a professional group, ward and professional experience.

## Results

Altogether 3,605 questionnaires were distributed, including 2,382 for nurses and 1,223 for physicians; 2,672 forms were returned, including 1,934 from nurses and 738 from doctors. The survey response rates reached 74%, from which those missing over 10% answers within the scale SAQ-SF (Safety Attitudes Questionnaire Short Form) were rejected. As a result, 1,133 questionnaires were finally analysed, including 606 from nurses and 527 from physicians. The majority were women (743; 65.57%). A similar percentage of nurses and physicians were employed in medical versus surgical wards– 50.33% vs. 49.67% nurses and 50.28% vs. 49.72%. In both professional groups, the largest percentage of staff worked in the wards where adult patients were treated (97.51% nurses vs. 99.33% physicians respectively). Over a half of nurses (n = 316; 54.02%) had been working for at least 21 years. The largest percentage of physicians had been working for 11 to 20 years (n = 133; 25.88%) or over 21 years (n = 127; 24.71%).

Statistically significant differences associated with gender and seniority were found between the group of nurses and physicians (p<0.05). A borderline difference was found between the nurses and physicians groups in relation to the age of the treated patients (p = 0.048), Table 1.

A multi-dimensional analysis showed that the teamwork climate (TC), safety climate (SC), job satisfaction (JS), stress recognition (SR), perception of management (PM) and work conditioning (WC) vary between the professional groups (nurses vs. physicians, F (6.1126) = 18.08, p<0.001), type of ward (medical vs. surgical, F (6.1126) = 3.49, p = 0.002), gender, F (6.1059) = 9.17, p<0.001) and seniority, F (24,3800) = 2.13, p = 0.001).

Both nurses and physicians received the highest mean results on the stress recognition subscale (SR) (71.6 and 80.86), while both groups evaluated work conditions (WC) the lowest (45.82 and 52.09). The analysis showed statistically significant differences between the groups of nurses and physicians in all 6 subscales, delineating different aspects of the evaluation of attitudes towards factors fostering patients' safety. The results of nurses in those subscales were lower compared to physicians, Table 2.

The staff working in surgical wards obtained higher mean results in stress recognition compared with the staff working in medical units (78.12 vs. 73.72; p = 0.001). Analyses found a statistically significant difference between men's and women's responses within all the six

**Table 1. Characteristics of the studied group.**

| | | nurses | physicians | |
|---|---|---|---|---|
| | | (N = 606) | (N = 527) | |
| | | N (%) | N (%) | p |
| **Unit type, n (%)** | | | | |
| | **Medical units** | 305 (50.33) | 265 (50.28) | 1.000 |
| | **Surgical units** | 301 (49.67) | 262 (49.72) | |
| **Gender, n (%)** | | | | |
| | **Women** | 553 (95.34) | 190 (39.09) | **0.000** |
| | **Men** | 27 (4.66) | 296 (60.91) | |
| **Age group of treated patients, n (%)** | | | | |
| | **Adults** | 509 (97.51) | 447 (99.33) | **0.048** |
| | **Adults & children** | 13 (2.49) | 3 (0.67) | |
| **Seniority, n (%)** | | | | |
| | **< 1 year** | 33 (5.64) | 36 (7) | **0.000** |
| | **1–4 years** | 65 (11.11) | 112 (21.79) | |
| | **5–10 years** | 74 (12.65) | 106 (20.62) | |
| | **11–20 years** | 97 (16.58) | 133 (25.88) | |
| | **≥21 years** | 316 (54.02) | 127 (24.71) | |

Note: p -value; N- number.

subscales, with women scoring significantly lower–$p < 0.05$. A statistically significant difference was shown between the professional experience of the surveyed staff and stress recognition (SR) ($p = 0.002$). Persons of lower seniority scored higher in the SR subscale, as shown in Table 2.

The largest percentage of both nurses and physicians showed positive attitudes (score ≥ 75 points) in relation to the safety of patients, towards stress recognition (39.1% and 61.5% respectively) and job satisfaction (21.5% and 36.4%). Work conditions received the lowest scores (5.8% and 13.1%). Except for stress recognition (SR), in other subscales, positive attitudes (≥75 points) were reported by fewer than 40% of the surveyed nurses and physicians. In all six subscales, a significantly higher percentage of physicians than nurses reported positive attitudes towards safety ($p < 0.05$), Table 3.

Statistical analysis further showed that each increase in the number of beds over the bed number contracted with the National Health Fund (Polish: NFZ) by 10 was connected with the increase in stress recognition by 2.5 units ($B^*(SE) = 2.5(0.7)$; $p = 0.001$) and the decrease in job satisfaction by about 1.6 points ($B^*(SE) = -1.6(0.8)$; $p = 0.030$).

## Discussion

The study is the first one to capture nurses' and physicians' attitudes towards patient safety in Polish hospitals based on the standardized version of the SAQ. In the five out of six SAQ subscales, a positive result, i.e. 75 pts. or more, was obtained by less than 40% of surveyed nurses and physicians. That finding suggests there is a need for significant investments in developing proactive patient safety cultures across Polish hospitals.

When comparing the findings internationally, the analyses showed that Polish nurses received lower mean results in the SR subscale than nurses working in Australia [9] or in Norway [10], but higher than nurses working in Sweden or Albania [11, 12]. Polish nurses also obtained higher results than nurses working in Asian countries (China, Turkey, Saudi Arabia, Iran) [13–16], African (Kenya) [17] or in Americas (Brazil and Pittsburgh) [18, 19] (Table 4).

**Table 2. Comparison of Safety Attitudes Questionnaire (SAQ-SF) results in reference to socio-demographic features.**

| | | Teamwork climate | Safety climate | Job satisfaction | Stress recognition | Perception of management | Work conditions |
|---|---|---|---|---|---|---|---|
| **Group** | | Mean (sd) | Mean (sd) | Mean (sd) | Mean (sd) | Mean (sd) | Mean (sd) |
| **Professional group, n (%)** | | | | | | | |
| | **Nurses (N = 606)** | 62.38 (17.53) | 63.41 (17.8) | 60.95 (22.21) | 71.6 (22.09) | 52.33 (22.03) | 45.82 (21.33) |
| | **Physicians (N = 527)** | 66.76 (15.57) | 66.06 (16.56) | 68.5 (22.7) | 80.86 (22.63) | 59.47 (21.61) | 52.09 (22.97) |
| | stat | $F_{(1,1131)} = 19.6$, $p < 0.001$ | $F_{(1,1131)} = 6.68$, $p = 0.01$ | $F_{(1,1131)} = 31.9$, $p < 0.001$ | $F_{(1,1131)} = 48.4$, $p < 0.001$ | $F_{(1,1131)} = 30.1$, $p < 0.001$ | $F_{(1,1131)} = 22.7$, $p < 0.001$ |
| **Gender, n (%)** | | | | | | | |
| | **Women (N = 743)** | 62.92 (16.94) | 63.65 (17.55) | 62.21 (22.71) | 74.09 (22.66) | 53.88 (22.48) | 46.74 (21.54) |
| | **Men (N = 323)** | 67.86 (15.52) | 66.56 (15.99) | 69.8 (21.58) | 79.83 (22.33) | 59.75 (20.58) | 52.9 (22.84) |
| | stat | $F_{(1,1064)} = 20.1$, $p < 0.001$ | $F_{(1,1064)} = 6.54$, $p = 0.011$ | $F_{(1,1064)} = 25.9$, $p < 0.001$ | $F_{(1,1064)} = 14.6$, $p < 0.001$ | $F_{(1,1064)} = 16.1$, $p < 0.001$ | $F_{(1,1064)} = 17.7$, $p < 0.001$ |
| **Unit type, n (%)** | | | | | | | |
| | **Medical units (570)** | 63.64 (16.15) | 64.66 (16.61) | 64.97 (22.42) | 73.72 (23.96) | 54.91 (22.72) | 48.25 (22.06) |
| | **Surgical units (563)** | 65.2 (17.39) | 64.63 (17.95) | 63.94 (23.07) | 78.12 (21.37) | 56.4 (21.47) | 49.23 (22.59) |
| | stat | $F_{(1,1131)} = 2.46$, $p = 0.117$ | $F_{(1,1131)} = 0$, $p = 0.976$ | $F_{(1,1131)} = 0.58$, $p = 0.447$ | $F_{(1,1131)} = 10.6$, $p = 0.001$ | $F_{(1,1131)} = 1.28$, $p = 0.258$ | $F_{(1,1131)} = 0.54$, $p = 0.461$ |
| **Group of treated patients, n (%)** | | | | | | | |
| | **Adults (N = 956)** | 64.56 (16.57) | 64.86 (17.09) | 64.66 (22.52) | 75.77 (22.85) | 55.69 (22.28) | 48.17 (22.03) |
| | **Adults & children (N = 16)** | 64.22 (22.98) | 61.53 (22.14) | 65.7 (23.14) | 72.27 (19.49) | 50.45 (17.39) | 57.42 (19.53) |
| | stat | $F_{(1,970)} = 0.01$, $p = 0.935$ | $F_{(1,970)} = 0.59$, $p = 0.443$ | $F_{(1,970)} = 0.03$, $p = 0.854$ | $F_{(1,970)} = 0.37$, $p = 0.542$ | $F_{(1,970)} = 0.88$, $p = 0.35$ | $F_{(1,970)} = 2.79$, $p = 0.095$ |
| **Seniority, n (%)** | | | | | | | |
| | **< 1 year (N = 69)** | 66.49 (15.71) | 65.52 (14.43) | 65.98 (20.82) | 78.99 (19.09) | 62.62 (19.55) | 50.63 (23.66) |
| | **1–4 years (N = 177)** | 63.77 (16.3) | 63.09 (16.72) | 61.84 (23.84) | 78.58 (21.19) | 55.52 (22.45) | 51.45 (23.13) |
| | **5–10 years (N = 180)** | 62.68 (17.21) | 62.88 (16.99) | 64.13 (22.86) | 79.54 (21.76) | 54.48 (22.7) | 46.84 (21.22) |
| | **11–20 years (N = 230)** | 63.53 (16.9) | 64.91 (17.35) | 65.07 (24.1) | 75.86 (24.88) | 55.63 (21.39) | 47.83 (22.27) |
| | **≥21 years (N = 443)** | 65.44 (16.76) | 65.36 (17.93) | 65.12 (22) | 72.74 (22.9) | 54.98 (22.43) | 48.6 (22.32) |
| | stat | $F_{(4,1094)} = 1.39$, $p = 0.235$ | $F_{(4,1094)} = 1.06$, $p = 0.374$ | $F_{(4,1094)} = 0.8$, $p = 0.523$ | $F_{(4,1094)} = 4.24$, $p = 0.002$ | $F_{(4,1094)} = 1.95$, $p = 0.1$ | $F_{(4,1094)} = 1.2$, $p = 0.31$ |

Note: sd- Standard deviation; stat– statistics; p -value.

Seniority or years of work experience also influenced the findings in ways consistent with international studies. According to this research, nurses with lower seniority demonstrated a better awareness of the negative impact of stress on patient safety. These results are similar to the results obtained by Aljadhey et al. [15], which documented that the increase of seniority corresponds to the decrease in stress recognition. The research by Żuralska et al. further showed that in difficult situations, nurses of lower seniority implemented a style more frequently focused on the task and seeking support than nurses of higher seniority. Nurses with fewer years of service took more cognitive and behavioural effort in order to manage stressful situations [24].

The results of the research by Rasool et al. [25] and Ganndi et al. [26] indicate that professional stress is a crucial element negatively influencing safety and work efficiency. When comparing physician specific findings to international results, Polish physicians participating in

**Table 3. Comparison of Safety Attitudes Questionnaire (SAQ-SF) results among nurses and physicians.**

| Teamwork climate (TC) | Category* | Nurses (N = 606) | | Physicians (N = 527) | | p |
|---|---|---|---|---|---|---|
| | | n | % | n | % | |
| | ≥ 75 | 116 | 19.1 | 134 | 25.4 | **0.013** |
| | < 75 | 490 | 80.9 | 393 | 74.6 | |
| **Safety climate (SC)** | ≥ 75 | 133 | 21.9 | 148 | 28.1 | **0.021** |
| | < 75 | 473 | 78.1 | 379 | 71.9 | |
| **Job satisfaction (JS)** | ≥75 | 130 | 21.5 | 192 | 36.4 | **0.000** |
| | < 75 | 476 | 78.5 | 335 | 63.6 | |
| **Stress recognition (SR)** | ≥ 75 | 237 | 39.1 | 324 | 61.5 | **0.000** |
| | < 75 | 369 | 60.9 | 203 | 38.5 | |
| **Perception of management (PM)** | ≥ 75 | 76 | 12.5 | 106 | 20.1 | **0.001** |
| | < 75 | 530 | 87.5 | 421 | 79.9 | |
| **Work conditions (WC)** | ≥ 75 | 35 | 5.8 | 69 | 13.1 | **0.000** |
| | < 75 | 571 | 94.2 | 458 | 86.9 | |

*≥75 pts.

–positive result. <75 pts.- negative result.

Note: p -value; N- number.

**Table 4. Compilation of study results on nurses' and physicians' attitudes towards factors conditioning safety of patients treated in hospitals.**

| | COUNTRIES | Teamwork Climate | Safety Climate | Job Satisfaction | Stress Recognition | Perceptions of Management | Working Conditions |
|---|---|---|---|---|---|---|---|
| | | **NURSES** | | | | | |
| **EUROPE** | **Poland n = 606** | 62.38 | 63.41 | 60.95 | 71.6 | 52.33 | 45.82 |
| | Norway n = 73 [10] | 75.0 | 70.6 | 80.4 | 78.0 | - | 72.7 |
| | Sweden n = 80 [11] | 67.2–68.8 | 62.3–63.3 | 70.6–72.0 | 66.1–69.9 | 48.2–48.5 | 55.6–59.5 |
| | Albania n = 132 [12] | 45.7 | 36.8 | 40.6 | 46.7 | 44.8 | 29.2 |
| **ASIA** | China n = 271 [13] | 73.9 | 69.2 | 74.0 | 59.0 | 67.8 | 73.5 |
| | Turkey n = 89 [14] | 46.27 | 46.44 | 58.91 | 65.12 | 46.83 | 51.93 |
| | Saudi Arabia n = 418 [15] | 75.5 | 75.5 | 92.7 | 41.9 | 68.1 | 82.1 |
| | Iran n = 244 [16] | 74.6 | 67.97 | 68.27 | 71.22 | 61.82 | 65.62 |
| **AFRICA** | Kenya n = 122 [17] | 72.0 | 65.7 | 77.7 | 56.9 | 63.5 | 55.9 |
| **AMERICA** | Brazil n = 46 [18] | 80.82 | 80.53 | 86.41 | 64.39 | 68.54 | 73.41 |
| | US, Pittsburgh n = 1,828 [19] | 72.5 | 72.8 | 71.3 | 68.8 | 59.9 | 66.7 |
| **AUSTRALIA** | Australia, Canberra n = 27 [9] | 75.62 | 70.60 | 74.07 | 79.86 | 51.54 | 50.69 |
| | | **PHYSICIANS** | | | | | |
| **EUROPE** | **Poland n = 527** | 66.76 | 66.06 | 68.5 | 80.86 | 59.47 | 52.09 |
| | Sweden n = 119 [11] | 69.2–72.2 | 53.5–60.6 | 68.8–69.8 | 68.7–76.2 | 49.3–56.0 | 52.5–57.6 |
| | Albania n = 209 [12] | 52.3 | 38.7 | 39.5 | 49.7 | 46.8 | 42.4 |
| **ASIA** | China n = 250 [13] | 75.5 | 71.1 | 75.7 | 65.8 | 69.7 | 71.4 |
| **AFRICA** | Kenya n = 49 [17] | 68.8 | 65.3 | 80.6 | 45.3 | 67.6 | 60.7 |
| **AMERICA** | US, Pittsburgh n = 1,352 [19] | 81.2 | 74.6 | 80.7 | 69.1. | 71.6 | 73.6 |
| **AUSTRALIA** | Australia. Canberra n = 24 [9] | 89.41 | 82.89 | 82.08 | 84.90 | 46.52 | 62.24 |

this study reported the highest mean value of the 6 subscales for stress recognition (SR) (80.86 points), which was similar to Australian doctors (84.90 points); however, it was higher than reported by Polish nurses (71,6 pts). This finding suggests that stress reduction measures for Polish physicians would be an important component for improving safety culture.

Effective teamwork can ameliorate or contribute to workplace associated stressors in ways that affect patient safety. In this study, Polish nurses received higher mean score in the teamwork subscale (TC) (62.38 pts.) than nurses working in Albania (45.7 pts.) [12] and Turkey (46.27 pts.) [14]. These results, however, were much lower compared to European countries such as Norway and Sweden [10, 11], Asia nations (China, Saudi Arabia, Iran) [13, 15, 16], countries in the Americas (Brazil, and US-PA, Pittsburgh) [18, 19], Australia [9] or Sub-Saharan Africa (Kenya) [17]. Polish physicians scored higher mean values in the area of team work (TC) than physicians from Albania [12], while lower than physicians working in Sweden, China, Kenya, America and Australia [9, 11, 13, 17].

The teamwork results could be attributed to the fact that physicians are prioritised in the hierarchy of the Polish healthcare system. Hierarchies impact teamwork dynamics and may prevent equal contribution of skills and knowledge of team members who are not doctors. Effective cooperation of various professionals in a team constitutes an important element of care and their patient safety.

In the assessment of safety climate (SC), Polish nurses received higher mean results (63.41 pts.) than nurses working in Albania (36.8 pts.) [12] and Turkey (46.44 pts.) [14], but definitely lower mean values than nurses working in Norway or Sweden [10, 11], as well as in the area of Asia (China, Saudi Arabia, Iran) [13, 15, 16], Americas (Brazil, and the state of Pennsylvania US-Pittsburgh) [18, 19], Australia (Australia) [9] or Africa (Kenya) [17]. Within the safety climate (SC) subscale, Polish physicians obtained higher mean results compared with the physicians from Sweden and Albania [11, 12], (Table 4).

Our study showed that nurses received a mean result of 60.95 pts. in the area of job satisfaction (JS), while physicians scored 68.5 pts—a difference that was highly significant. This result indicated that although nurses and physicians overall liked working in this hospital, nevertheless, nurses job satisfaction was significantly lower, which may affect patient safety. Kunaviktikul et al. [27] found that nurses who are not satisfied with their job tend to commit medical errors and that improvement in working conditions—including teamwork—is an important element to increase job satisfaction [27].

The results are consistent with the results of other authors, who showed that both nurses and physicians' evaluation of hospital management (PM) and work conditions (WC) is low from the perspective of their impact on the safety of treated patients [9, 11, 19]. Low scores in this subscale may be interpreted as the perception of managerial weakness in the context of patients' safety. According to Alayed et al. [28] the evaluation of management may be low due to the lack of direct contacts of employees with the managing staff or the lack of perception of the supervisor's engagement. Authors of other studies indicated that regular audits of staff managing healthcare facilities, common analysis of threats to patients' safety in wards, along with the provision of resources and support from supervisors were related to a very positive attitude of the employees to the factors conditioning patients safety [29, 30].

Our study also showed that 87% of nurses (530/606) and almost 80% of physicians (421/527) gave low ratings to work conditions (WC) on patient safety. Similarly, low ratings of working conditions on patients' safety was also observed in other countries [9, 11, 12, 14, 16, 17, 31, 32] (Table 4). The results of our study within the subscale WC were lower than the data obtained from nurses employed in Norway, Sweden or Brazil [33, 34]. The analysis of the OECD (Organisation for Economic Cooperation and Development) data, indicated that in a country with a higher nurse rate (number of nursing working per 1,000 patients) working

conditions (WC) were scored higher (Table 4) [33]. The analysis of data from the OECD presenting the rate of nurses and doctors per 1,000 patients in particular countries indicates that in a country with a higher nurse rate work conditions (WC) were scored higher.

Staffing resources, therefore, are an important component for building capacity to foster patient safety systems and cultures. For Poland, the rate of nurses per 1,000 patients is 5.1, while in Norway—18.0, in the USA and Australia—11.9, in Brazil—10.11, in Sweden—10.9 [33, 34]. In reference to our study, the above data may be treated as a signal of a great need to increase the number of nurses in medical and surgical wards in order to optimise patients' safety hospitalised in the wards. In the RN4CAST study, the connection between the nurse staffing, education and hospital mortality was documented [35]. Research by Aiken et al. further indicated that the increase of workload for nurses by one patient increased the risk of the patient's death within 30 days of admission to the hospital [35]. Hence, evidence suggests that improved working conditions and increased nurse staffing leads to patient safety improvements.

Other studies by Wagner et al. and McHugh et al. showed that working in good conditions correlates to greater satisfaction and motivation to work, lower stress level among employees and higher patients' safety [36, 37]. Under Polish conditions, the present results indicate the need for management actions aimed at improving working conditions and reducing work stress by the hospital. It is also necessary to improve communication and cooperation within the treatment team, recognizing the competencies and responsibilities of individual members. It can be expected that the above actions will contribute to the improvement (attitude) towards safety among nurses and physicians employed in the hospital.

The cohesion of the results concerning nurses' and physicians' attitudes towards patient safety conditions of those treated in the medical and surgical wards in hospitals in Poland and in other culturally and geographically differing countries points to a universal character of the patient safety cultures. The recently introduced Global Patient Safety Action Plan: 2021–2030, which includes broadly developed strategies that can be adapted to the national context, can be used to do this [38]. Their implementation will help ensure that patient safety is improved both nationally and globally. Studies like this one are a first step toward capturing the necessary baseline data from which positive changes can be made.

## Research limitations

The study has some limitations. Firstly, the data were collected exclusively from surgical and medical wards. That is why direct results of the study are limited to the specific conditions of the wards studied, and one has to be careful before generalisation for other types of hospital wards. Secondly, the research was voluntary. The 21 out of 949 hospitals located all over the country had directors and staff who agreed to allow the research in their organization. Subsequent future studies should expand research to a larger number of hospitals and different types of wards (including pediatric, intensive care and psychiatric) with additional inclusion of the influence of factors conditioning attitudes towards safety on the frequency of adverse events, including international studies comparing different countries. In addition, in subsequent studies, to accurately assess the accuracy of the scale, it should also be tested by comparing different groups of medical professions, and their selection would take into account a comparable number of respondents.

## Conclusions

The results are valuable to identify areas for improvement related to patient safety, including at the unit level. Managers' awareness of the importance of coping with staff occupational

stress, working conditions and effective teamwork can contribute to improving employees' attitudes towards patient safety. The differences noted between countries are important because they demonstrate a consistency in the measurement of the SAQ tool across languages and health systems, thereby allowing for future comparative country studies and the potential for international benchmarking of patient safety culture. Overall, this study contributes to the growing body of literature that highlights how the conditions that foster patient safety during hospitalization can be consistent across any healthcare organization.

## Supporting information

**S1 Data.**
(XLSX)

**S2 Data.**
(XLSX)

## Acknowledgments

We would like to thank all who took part in in this study.

## Author Contributions

**Conceptualization:** Iwona Malinowska-Lipień.

**Data curation:** Iwona Malinowska-Lipień, Maria Kózka, Agnieszka Gniadek, Tomasz Brzostek.

**Formal analysis:** Iwona Malinowska-Lipień, Agnieszka Micek, Krzysztof Gajda, Agnieszka Gniadek, Tomasz Brzostek, Allison Squires.

**Funding acquisition:** Iwona Malinowska-Lipień.

**Investigation:** Iwona Malinowska-Lipień, Teresa Gabryś.

**Methodology:** Iwona Malinowska-Lipień, Agnieszka Micek, Teresa Gabryś, Maria Kózka, Krzysztof Gajda, Agnieszka Gniadek, Tomasz Brzostek.

**Project administration:** Iwona Malinowska-Lipień.

**Resources:** Iwona Malinowska-Lipień.

**Software:** Agnieszka Micek, Krzysztof Gajda.

**Supervision:** Teresa Gabryś, Tomasz Brzostek, Allison Squires.

**Validation:** Iwona Malinowska-Lipień.

**Visualization:** Iwona Malinowska-Lipień.

**Writing – original draft:** Iwona Malinowska-Lipień, Maria Kózka, Agnieszka Gniadek, Tomasz Brzostek.

**Writing – review & editing:** Iwona Malinowska-Lipień, Maria Kózka, Agnieszka Gniadek, Tomasz Brzostek, Allison Squires.

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
