## [Decision Letter · Decision Letter 0]

31 Aug 2021

PONE-D-21-20381

Attitudes of nurses and physicians towards factors affecting hospitalized patient safety

PLOS ONE

Dear Dr. Malinowska-Lipień,

Thank you for submitting your manuscript to PLOS ONE. After careful consideration, we feel that it has merit but does not fully meet PLOS ONE’s publication criteria as it currently stands. Therefore, we invite you to submit a revised version of the manuscript that addresses the points raised during the review process.

We look forward to receiving your revised manuscript.

Kind regards,

Sandul Yasobant, PhD

Academic Editor

PLOS ONE

Journal Requirements:

2. Please improve statistical reporting and report exact p-values for all values greater than or equal to 0.001, and refer to p-values as "p<.001" instead of "p=.000". Our statistical reporting guidelines are available at https://journals.plos.org/plosone/s/submission-guidelines#loc-statistical-reporting.

Reviewers' comments:

Reviewer's Responses to Questions

**Comments to the Author**

1. Is the manuscript technically sound, and do the data support the conclusions?

Reviewer #1: Yes

Reviewer #2: No

2. Has the statistical analysis been performed appropriately and rigorously? 

Reviewer #1: Yes

Reviewer #2: I Don't Know

3. Have the authors made all data underlying the findings in their manuscript fully available?

Reviewer #1: No

Reviewer #2: Yes

4. Is the manuscript presented in an intelligible fashion and written in standard English?

Reviewer #1: Yes

Reviewer #2: No

5. Review Comments to the Author

Reviewer #1: The manuscript is technically sound, in which the authors have been able to demonstrate a good command of existing literature; they have provided a detailed explanation of the methodology used including rigorous data analysis; the results section is well presented; discussion is done with sufficient details and robust literature that has compared the country context with that of other countries both high- income and low- and middle- income countries; and the limitations and conclusions are well described. This indicates that the manuscript is "presented in an intelligible fashion" and to the best of my knowledge, it is "written in standard English". The authors have included all the important data in the manuscript and they have indicated in their submission that in terms of data availability - "No - some restrictions will apply".

Additionally, regarding the content of the manuscript, the authors have done an excellent work that contribute a significant body of knowledge in this very important aspect of quality of care, namely “patient safety”. They have presented original research that was done in Poland involving 21 (out of 949) hospitals all over the country. The “introduction” section of the manuscript is well presented indicating good command of existing literature and building well on the justification for conducting the study. In this section, I suggest that the authors make a correction in the last sentence of the second paragraph on page 3 which reads: “The analysis of factors contributing to the occurrence of adverse events is underlying condition for changes in the staff's behaviour, which may makes healthcare environment safer [7];” by replacing the word “makes” with “make” so that it will read as follows: “The analysis of factors contributing to the occurrence of adverse events is underlying condition for changes in the staff's behaviour, which may make healthcare environment safer [7].

In the “materials and methods” section in the fourth paragraph on page 5, I suggest that the authors delete the word “towards” in the following sentence: “Its first part includes questions related to the following subscales:: 1/ Teamwork climate -…….; 2/Safety climate - SC (questions from 7 to 13) – evaluates the perception of employees' organisational involvement in towards patient’s safety; 3/Job satisfaction - JS (questions from 15 to 19) –……”, so that it will read: “……..of employees' organisational involvement in patient’s safety;” The statistical analysis is well described with adequate details.

The results section is well presented logically and in details.

The authors have done an excellent job in the discussion section by providing detailed discussion backed up with comprehensive literature comparing the Polish situation with that of other countries and in different contexts including a sub-Saharan African country (Kenya). This makes the discussion section robust and rich with data from high income and low- and middle-income countries. The paragraph 8 on pages 14-15, has corroborated the findings of a recent systematic review on patient safety culture, in terms of the importance of having adequate number of staff, effective communications, supervisor’s (management) support, and teamwork in building a “patient safety culture” [1]. The paragraph 9 points to the needs for using similar strategies in the various countries with different cultural and geographical contexts. In here, I suggest that the authors add a mention of also implementing the recently launched “Global Patient Safety Action Plan: 2021-2030”, which is broad and with well elaborated strategies that can be adapted into country context that will ensure that patient safety is improved at country and global level [2].

The “research limitations” section is well written pointing out the key limitations. The “conclusion” section is logically written based on the findings. Their emphasis on the need for managers to address “occupational stress” is also supported with the study by Park and Kim (2013) in Korea in which they found that “job stress” was one of the factors that affected “patient safety incidents” [3].

1. Reis CT, Paiva SG, Sousa P. The patient safety culture: a systematic review by characteristics of Hospital Survey on Patient Safety Culture dimensions. Int J Qual Health Care. 2018;30(9):660-677. DOI: 10.1093/intqhc/mzy080.

2. World Health Organization. Global patient safety action plan 2021–2030: towards eliminating avoidable harm in health care. ISBN 978-92-4-003270-5 (electronic version). Geneva: World Health Organization; 2021. Licence: CC BY-NC-SA 3.0 IGO. Available at: https://www.who.int/teams/integrated-health-services/patient-safety/policy/global-patient-safety-action-plan Accessed on 11th August, 2021.

3. Park YM, Kim SY. Impacts of Job Stress and Cognitive Failure on Patient Safety Incidents among Hospital Nurses. Saf Health Work. 2013;4(4):210-215. doi: doi: 10.1016/j.shaw.2013.10.003

Reviewer #2: - Spelling, grammar and English need detailed review and edition.

Title

- Better if you specify specific factors

- It looks like attitude towards factors not patient safety

- It is very confusing title

Abstract

- Key words seem like sentence

- Why medical and surgical nurses only

Introduction

- Why you want to study this and how you do it differently are not stated well.

Aim

- What about other specific objectives

Methods and materials

- Specify the validity and reliability of tool

- What do you do to improve the quality of your tool?

- How you select data collectors

- You need comprehensive literature search to include all independent variables

- The operational definition is not clear, please write it clearly and in detail

- What about inclusion and exclusion criteria?

- How you minimize confounding factors

Result, discussion and limitations

- How you minimize correlation

- It’s difficult to add up the number of nurses and doctors’ response, how do you do it?

- How do you classify the age group?

- Why surgical and medical ward only

- You need to explain the possible reasons in study differences and similarities

- Some of works you should do did not need to be stated as limitations

-

6. PLOS authors have the option to publish the peer review history of their article (what does this mean?). If published, this will include your full peer review and any attached files.

Reviewer #1: **Yes: **Eliudi Saria Eliakimu

Reviewer #2: No

---

## [Author Response · Author response to Decision Letter 0]

14 Oct 2021

Dear Reviewers,

The authors would like to thank the reviewer for his/her thorough review of the manuscript. We believe that this revised version, which includes reviewers’ suggestions, is more accurate and communicates better the main message of the article. 

We will gratefully respond to any further comments on the text.

The paragraph is marked in yellow in the manuscript.

Thank you again for taking the time to review our paper and for your constructive comments.

Following comments have been modified in a new version of the manuscript:

Yours sincerely,

IML

Reviewer(s)' Comments to Author: 

Reviewer: 1 

I suggest that the authors make a correction in the last sentence of the second paragraph on page 3 which reads: “The analysis of factors contributing to the occurrence of adverse events is underlying condition for changes in the staff's behaviour, which may makes healthcare environment safer [7];” by replacing the word “makes” with “make” so that it will read as follows: “The analysis of factors contributing to the occurrence of adverse events is underlying condition for changes in the staff's behaviour, which may make healthcare environment safer [7]. 

Authors' response:

We would like to thank the Reviewer for his due remark. Has been changed.

In the “materials and methods” section in the fourth paragraph on page 5, I suggest that the authors delete the word “towards” in the following sentence: “Its first part includes questions related to the following subscales:: 1/ Teamwork climate -…….; 2/Safety climate - SC (questions from 7 to 13) – evaluates the perception of employees' organisational involvement in towards patient’s safety; 3/Job satisfaction - JS (questions from 15 to 19) –……”, so that it will read: “……..of employees' organisational involvement in patient’s safety;” 

Authors' response:

We would like to thank the Reviewer for his due remark. Has been changed.

In here, I suggest that the authors add a mention of also implementing the recently launched “Global Patient Safety Action Plan: 2021-2030”.

https://wfsahq.org/news/latest-news/whas-global-patient-safety-plan-2021-30/

Authors' response:

We would like to thank the Reviewer for the suggestion to refer to the "Global Patient Safety Action Plan: 2021-2030" in this manuscript. Used in the text under “Discussion”, paragraph 9. 

We have included the suggestion as follows:.

The recently introduced Global Patient Safety Action Plan: 2021-2030, which includes broadly developed strategies that can be adapted to the national context, can be used to do this. Their implementation will help ensure that patient safety is improved both nationally and globally. Therefore, all the more a safety culture should permeate the attitudes, beliefs, values, skills and practices of healthcare professionals, managers and leaders of healthcare organizations

Reviewer: 2 

Spelling, grammar and English need detailed review and edition. 

Authors' response:

The text has been corrected.

Title

- Better if you specify specific factors

- It looks like attitude towards factors not patient safety

- It is very confusing title

Authors' response:

We would like to thank the Reviewer for his attention. We changed the manuscript title to: 

Nurses and physicians attitudes towards factors related to hospitalized patient safety

Abstract

- Key words seem like sentence

Authors' response:

We would like to thank the Reviewer for his correct remark, the key words have been changed and clarified to:

safety, attitudes, patients, nurses, physicians

- Why medical and surgical nurses only 

Authors' response:

The research was carried out according to the protocol of the RN4CAST project, which included the study of a nurse from the surgical and medical departments. The study group worked in 21 hospitals representing a representative sample for Poland, and selected according to the geographical area of the country, population density, taking into account the hospital's reference level. The study was cross-sectional and correlational. In line with the assumptions of the RN4Cast project, internal medicine and surgery departments were selected because they provide multidisciplinary care provide health care to the majority of urgently hospitalized adult patients, provide multidisciplinary care and employ a large number of nurses.

Introduction

- Why you want to study this and how you do it differently are not stated well. 

Authors' response:

This research was conducted in accordance with the RN4CAST project protocol. Replication the RN4CAST protocol could help to analyse developing a culture safety. In addition, the inclusion of a new short safety assessment tool (SAQ questionnaire - the new tool used for this study) it is useful in the assessment of patient safety determinants by managers. When shaping a safe culture in health care organization, it should be remembered that ensuring the safety of patient care is primarily human resources - staff involved in the treatment process - mainly doctors and nurses. High awareness of occupational safety, cooperation in an interdisciplinary team, assessment of the culture of work safety drawing conclusions and making the right decisions may will increase the safety of hospitalized patients. On the other hand, it should be remembered that a strong safety culture is not only key to reducing harm to patients, but is also key to ensuring a safe working environment for healthcare professionals, which is why it is so important to undertake research to assess attitudes towards safety.

Aim

- What about other specific objectives 

Authors' response:

Specific objectives were included in the manuscript. 

Detailed objectives: 

1. What attitudes towards factors related to hospitalized patient safety are presented by nurses and by doctors? 

2. What were the differences in attitudes towards safety in the group of nurses and doctors? 

3. To what extent did the type of ward differentiate the attitudes towards safety of nurses and doctors? 

4. What was the relationship between the respondents' gender and attitudes towards safety? 

5. What was the relationship between the time of employment of the respondents and their attitudes towards safety?

Methods and materials

- Specify the validity and reliability of tool

Authors' response:

As suggested by the Reviewer, the text of the manuscript has been supplemented in the "Material and methods" section, paragraph 3. 

The Safety Attitudes Questionnaire reliability had a Cronbach's Alpha of 0.98. Before performing the analysis validity of the Polish adaptation of SAQ-SF, the Kaiser test was used to check whether the data meet the requirements of the factor analysis. The Kaiser-Mayer-Olkin (KMO) value, being the measure of the adequacy of the sample selection, was estimated at the level of 0.87 (df=8630, p<0.001). This model explained 68% of the total variance of the analysed set of variables (Malinowska – Lipień, et al. 2021).

What do you do to improve the quality of your tool?

Authors' response:

The subsequent study is planned to improve the quality of the tool.

- How you select data collectors

Authors' response:

 At the hospital, a hospital coordinator was appointed to liaise with the research team and ensure consistency in data collection in line with the study guidelines. In clinical departments, the questionnaire was distributed among departmental staff by the hospital project coordinator. Participants had 4 weeks to complete the questionnaire. The completed anonymous questionnaires were deposited through an opening in securely closed boxes. Neither the coordinators nor anyone else at the hospital level had access to the contents of the boxes, they were only opened by a team of researchers. After 4 weeks, the coordinators collected the boxes, secured them and sent them by courier over to the research team. Before starting the study every participant was informed that participation was voluntary and anonymous, all responses are kept confidential and no individual responses would be available to the hospital management.

- You need comprehensive literature search to include all independent variables

Authors' response:

 The research results presented in the reviewed manuscript concern attitudes towards the patient safety of nurses and doctors assessed with the SAQ questionnaire. The independent variables were data obtained only from the SAQ questionnaire. The authors agree with the Reviewer that independent variables do not include all possible variables. There is no analysis regarding the age of the respondents. However, the questionnaire does not include such data. 

The analysis was carried out in accordance with the principles described in the refences:

10. Bondevik GT, Hofoss D, Husebø SB, Tveter Deilkås EC. Patient safety culture in Norwegian nursinghomes. BMC Health Services Research. 2017, 17:424. doi 10.1186/s12913-017-23.

11. Milton J, Chaboyer W, Åberg ND, Andersson AE, Oxelmark L. Safety attitudes and working climate after organizational change in a major emergency department in Sweden. Int Emerg Nurs. 2020, 3;100830. doi: 10.1016/j.ienj.2020.100830.

12. Gabrani A, Hoxha A, Simaku A, Gabrani J. Application of the Safety Attitudes Questionnaire (SAQ) in Albanian hospitals: a cross-sectional study. BMJ Open. 2015;5:e006528, doi: 10.1136/bmjopen-2014-006528.

13. Jiang K, Tian L, Yan C, Li Y, Fang H, Peihang S, Li P, Jia H, Wang Y, Kang Z, Cui Y, Liu H, Zhao S, Anastasia G, Jiao M, Wu Q, Liu M. A cross-sectional survey on patient safety culture in secondary hospitals of Northeast China. PLoS One. 2019, 14(3):e0213055. doi:10.1371/journal.pone.0213055.

14. Bahar S, Önler E. Turkish surgical nurses' attitudes related to patient safety: A questionnaire study. Niger J Clin Pract. 2020, 23(4):470-475. doi: 10.4103/njcp.njcp_677_18.

- The operational definition is not clear, please write it clearly and in detail

Authors' response:

 As suggested, the definition according to WHO and The International Classification for Patient Safety (ICPS) was included.

For the WHO, “ patient safety means the reduction to an acceptable minimum level of risk of unnecessary harm related to health care”. The International Classification for Patient Safety (ICPS) defines ‘patient safety’ “as the act of avoiding, preventing or improving adverse outcomes or injuries occurred throughout the medical-hospital process”

- What about inclusion and exclusion criteria?

Authors' response:

 As suggested by the Reviewer, the text of the manuscript has been supplemented in the "Material and methods" section, paragraph 2. 

The inclusion criterion was employment and active work during the study as a nurse or physician in a given hospital. All other hospital staff was excluded from the study. Nurses on maternity leave, extended

sick leave or study leave were excluded from participation.

- How you minimize confounding factors 

Authors' response:

Potential confusing factors have been limited by through clear written instructions attached to the each questionnaire, and also by detailed familiarization of hospital coordinators with the meaning of the study and method of filling in forms, which was then handed over to participants. The research team provided necessary additional information and monitored the process of obtaining completed forms.

Result, discussion and limitations

- How you minimize correlation

Authors' response:

 Comparison of teamwork climate (TC), safety climate (SC), job satisfaction (JS), recognition of stress (SR), management perception (PM), working conditions (WC) between occupational groups, types of departments, genders, age groups patients covered by the care and seniority of nurses and doctors was first carried out using MANOVA - to take into account the influence of these factors on all aspects of safety attitudes at the same time, and taking into account the correlations between them.

- It’s difficult to add up the number of nurses and doctors’ response, how do you do it? 

Authors' response:

For some analyzes (e.g. unit type, gender, and work experience), the nurses 'and physicians' scores were summed up as both occupational groups completed the same SAQ. Both the questions of the questionnaire and the calculation method are the same for both occupational groups.

- How do you classify the age group?

Authors' response:

 In the studies, the study group was not divided by age, but only by work experience and the group of patients treated in the study unit (adults and adults & children). The time frame of work experience and the division into a group of patients are consistent with the original version of the SAQ. 

- Why surgical and medical ward only 

Authors' response:

Studies for this manuscript were performed according to the RN4CAST project protocol. This study was carried out in a group of nurses working in departments, surgical and internal medicine departments, in 21 hospitals with 24-hour permanent duty, representative a representative sample for Poland, and selected according to the geographical area of the country, population density taking into account the hospital's reference level. The study was cross-sectional and correlational. In line with the RN4Cast project, the medical and surgical departments were selected because they provide multidisciplinary care and employ a large number of nurses.

- You need to explain the possible reasons in study differences and similarities 

Authors' response:

Included in the manuscript. 

Polish nurses received higher mean score in the teamwork subscale mean results in the subscale concerning teamwork (TC) (62.38 pts.) than nurses working in Albania (45.7 pts.) [12] and Turkey (46.27 pkt.) [14], but critically lower compared to European countries as Norway and Sweden [10-11], Asia (China, Saudi Arabia, Iran) [13, 15-16], Americas (Brazil, and US-PA, Pittsburgh) [18-19], Australia [9] or Africa (Kenya) [17]. The evaluation of team work by Polish nurses may result from the fact that position of physician is being prioritised in the hierarchy of the Polish healthcare system. This opinion has an impact on the teamwork and may prevent equal contribution of skills and knowledge of from other health team members those team members who are not doctors. Effective cooperation of various professionals in a team constitutes an important element of care and their patient safety.

In the aassessment of safety climate (SC), Polish nurses received higher mean results (63.41 pts.) than nurses working in Albania (36.8 pts.) [12] and Turkey (46.44 pkt.) [14], but definitely lower mean values than nurses working in Norway or Sweden [10-11], as well as in the area of Asia (China, Saudi Arabia, Iran) [13, 15-16], Americas (Brazil, and the state of Pennsylvania US-Pittsburgh) [18-19], Australia (Australia) [9] or Africa (Kenya) [17]. Within the safety climate (SC) subscale, Polish physicians obtained higher mean results compared with the physicians doctors from Sweden and Albania [11-12]

- Some of works you should do did not need to be stated as limitations 

Authors' response:

The presented limitations of the work result from other assumptions of the presented research and the lack of available data. The authors consider these issues to be important to consider in future research (e.g., different types of wards, more hospitals, different medical professions). The authors believed that these were more limitations rather than the results of the research performed. Limitations, the reduction of which in subsequent studies will allow for a better understanding of the factors that determine the patient's safety.

---

## [Decision Letter · Decision Letter 1]

25 Oct 2021

PONE-D-21-20381R1Nurses and physicians attitudes towards factors related to hospitalized patient safetyPLOS ONE

Dear Dr. Malinowska-Lipień,

Thank you for submitting your manuscript to PLOS ONE. After careful consideration, we feel that it has merit but does not fully meet PLOS ONE’s publication criteria as it currently stands. Therefore, we invite you to submit a revised version of the manuscript that addresses the points raised during the review process.

 Dear Authors,The comments from Reviewer-2 is well received and valid points. Therefore, may I request you to kindly strengthen the method section especially quantifying the outcome and mentioning the confounding factors, perhaps in the limitation. Also my request you to kindly do language edits before you submit the next version.Thanks 

We look forward to receiving your revised manuscript.

Kind regards,

Sandul Yasobant, PhD

Academic Editor

PLOS ONE

Journal Requirements:

Reviewers' comments:

Reviewer's Responses to Questions

**Comments to the Author**

1. If the authors have adequately addressed your comments raised in a previous round of review and you feel that this manuscript is now acceptable for publication, you may indicate that here to bypass the “Comments to the Author” section, enter your conflict of interest statement in the “Confidential to Editor” section, and submit your "Accept" recommendation.

Reviewer #1: All comments have been addressed

Reviewer #2: (No Response)

2. Is the manuscript technically sound, and do the data support the conclusions?

Reviewer #1: Yes

Reviewer #2: Yes

3. Has the statistical analysis been performed appropriately and rigorously? 

Reviewer #1: Yes

Reviewer #2: Yes

4. Have the authors made all data underlying the findings in their manuscript fully available?

Reviewer #1: No

Reviewer #2: No

5. Is the manuscript presented in an intelligible fashion and written in standard English?

Reviewer #1: Yes

Reviewer #2: No

6. Review Comments to the Author

Reviewer #1: One typing error for correction in the fourth (04th) paragraph of “Discussion section” - the first sentence, which reads: “In the aassessment of safety climate (SC), Polish nurses received…….”. Spellings for the word assessment need to be corrected.

Reviewer #2: - Thank you for addressing most of the comments

- Still your English and grammar need detailed review and edition

Title

- Why nurses and doctors were your priority? What about other health professionals?

- “Nurses and physicians’ attitudes towards factors related to hospitalized patient safety”, please specify what factor you intend to study?

- “What about other staff nurses…” need further explanation?

Methods

- Operationalize patient safety, attitude in quantitative way (quantify it)

- How you minimize confounding factors needs further explanation

- Which age group is adult and which one is children

7. PLOS authors have the option to publish the peer review history of their article (what does this mean?). If published, this will include your full peer review and any attached files.

Reviewer #1: **Yes: **Eliudi Saria Eliakimu

Reviewer #2: No

---

## [Author Response · Author response to Decision Letter 1]

2 Nov 2021

Dear Reviewers, 

The authors would like to thank the reviewer for his/her thorough review of the manuscript. We believe that this revised version, which includes reviewers’ suggestions, is more accurate and communicates better the main message of the article. 

We will gratefully respond to any further comments on the text.

The paragraph is marked in green (English and grammar) and yellow (other changes) o in the manuscript.

Thank you again for taking the time to review our paper and for your constructive comments.

Best regards,

Reviewer #1: 

One typing error for correction in the fourth (04th) paragraph of “Discussion section” - the first sentence, which reads: “In the aassessment of safety climate (SC), Polish nurses received…….”. Spellings for the word assessment need to be corrected.

Response: The text was checked in terms of language and grammar. The corrected text is highlighted in green in the manuscript.

Reviewer #2: 

- Still your English and grammar need detailed review and edition

Response: The text was checked in terms of language and grammar. The corrected text is highlighted in green in the manuscript.

 Title 

- Why nurses and doctors were your priority? What about other health professionals?

Response: In Poland, the largest group of medical professions are nurses and doctors. It is doctors and nurses who play the main role in the process of diagnosis and treatment and patient care. According to the report of the National Health Fund, he works in hospitals (https://ezdrowie.gov.pl/portal/home/zdrowe-dane/zecja/zecja-dotyczace-liczby-personelu-medycznego-w-umowach-z-narodowym-funduszem-zdrowia ):

nurses - 192 223 

doctors - 113 724 

physiotherapists - 38 416 

midwives - 25 033 

laboratory diagnosticians - 760 

pharmacists - 18 564 

In the "Research limitations" part, the question of extending the study group to include representatives of other medical professions was taken into account. Therefore: "In addition, in subsequent studies, to accurately assess the accuracy of the scale, it should also be tested by comparing different groups of medical professions, and their selection would take into account a comparable number of respondents."

- “Nurses and physicians’ attitudes towards factors related to hospitalized patient safety”, please specify what factor you intend to study?

Response: The research used the international SAQ scale, which allows the assessment of patient safety in 6 areas: 1/Teamwork Climate; 2/ Safety Climate; 3/ Job Satisfaction; 4/Stress Recognition; 5/Perceptions of Management; 6/ Working Conditions. The research was carried out on the basis of the SAQ questionnaire adapted to Polish conditions (Malinowska-Lipień I, Brzyski P, Gabryś T, Gniadek A, Kózka M, Kawalec P, et al. Cultural adaptation of the Safety Attitudes Questionnaire - Short Form (SAQ-SF) in Poland. PLoS One. 2021 Feb 5;16(2):e0246340. doi: 10.1371/journal.pone.0246340).

- “What about other staff nurses…” need further explanation? 

Response: In Poland, according to the report of the Ministry of Health on a national scale, the largest number of beds was intended for patients requiring surgical treatment - 39.3 thousand. and for internal medicine patients - 23.0 thousand (Report - pp. 44 - 45, chart 18 (English version), therefore it was decided that the research should focus on this group of nurses and doctors. Of course, it is a correct remark that in subsequent studies it is necessary to include staff working in all types of wards, this was taken into account parts in "Research limitations" (Report: https://stat.gov.pl/files/gfx/portalinformacyjny/pl/defaultaktualnosci/5513/1/9/1/zdrowie_i_ochrona_zdrowia_w_2018.pdf)

“Subsequent future studies should expand research to a larger number of hospitals and different types of wards (including pediatric, intensive care and psychiatric) with additional inclusion of the influence of factors conditioning attitudes towards safety on the frequency of adverse events, including international studies comparing different countries”.

Methods 

- Operationalize patient safety, attitude in quantitative way (quantify it)

Response: The SAQ results were presented both as quantitative (Table 1) and qualitative (Table 2) data. The results of own research and quantitative data were compared in the discussion with the results of research carried out in 11 different countries. In all these studies, quantitative data were presented, our results were additionally enriched with qualitative data presented in Table 3. The results of the individual 6 subscales were divided into> = 75 points and <75 points, where * ≥75 pts. - positive result. <75 pts.- negative result.

- How you minimize confounding factors needs further explanation 

Response: The "Materials and methods" section describes the data collection process in detail, ensuring such conditions minimized disruptive factors.

At the hospital, a hospital coordinator has been appointed to work with the research team and ensure consistency in data collection in line with the guidelines of the study. In clinical departments, the questionnaire with a plastic envelope attached to it was distributed during departmental staff meetings led by the hospital project coordinator. Participants had 4 weeks to complete the questionnaire. Completed questionnaires packed in a sealed plastic envelope were thrown into a sealed box with a hole. After 4 weeks, the coordinator was responsible for collecting the boxes, securing them and handing them over to the research team. Participants were informed that participation was voluntary and anonymous, that all responses would be kept confidential and that no individual responses would be available to hospital management.

- Which age group is adult and which one is children

Response: The SAQ questionnaire in the "BACKGROUND INFORMATION" section has a question about the majority of patients in the ward. There are three distractors 1- adults; 2 kids; 3- both groups (original version of the SAQ: https://med.uth.edu/chqs/wp-content/uploads/sites/75/2020/03/SAQ-Short-Form-2006.pdf ; Polish version of SAQ: https://journals.plos.org/plosone/article?id=10.1371/journal.pone.0246340 ). In Poland, adult departments cover patients from 18 years of age, and children's departments up to 18 years of age.

---

## [Decision Letter · Decision Letter 2]

8 Nov 2021

PONE-D-21-20381R2Nurses and physicians attitudes towards factors related to hospitalized patient safetyPLOS ONE

Dear Dr. Malinowska-Lipień,

Thank you for submitting your manuscript to PLOS ONE. After careful consideration, we feel that it has merit but does not fully meet PLOS ONE’s publication criteria as it currently stands. Therefore, we invite you to submit a revised version of the manuscript that addresses the points raised during the review process.

Dear Authors,The current draft still requires English language editing, as correctly pointed out by one of our reviewers. Please re-submit it with help of a professional proof reader. Thanks

We look forward to receiving your revised manuscript.

Kind regards,

Sandul Yasobant, PhD

Academic Editor

PLOS ONE

Journal Requirements:

Reviewers' comments:

Reviewer's Responses to Questions

**Comments to the Author**

1. If the authors have adequately addressed your comments raised in a previous round of review and you feel that this manuscript is now acceptable for publication, you may indicate that here to bypass the “Comments to the Author” section, enter your conflict of interest statement in the “Confidential to Editor” section, and submit your "Accept" recommendation.

Reviewer #2: All comments have been addressed

2. Is the manuscript technically sound, and do the data support the conclusions?

Reviewer #2: Partly

3. Has the statistical analysis been performed appropriately and rigorously? 

Reviewer #2: Yes

4. Have the authors made all data underlying the findings in their manuscript fully available?

Reviewer #2: Yes

5. Is the manuscript presented in an intelligible fashion and written in standard English?

Reviewer #2: Yes

6. Review Comments to the Author

Reviewer #2: still it needs to be edited with english editor. Grammar and preposition needs to be properly addresed.

7. PLOS authors have the option to publish the peer review history of their article (what does this mean?). If published, this will include your full peer review and any attached files.

Reviewer #2: No

---

## [Author Response · Author response to Decision Letter 2]

16 Nov 2021

Dear Reviewer, 

The authors would like to thank the reviewer for review of the manuscript. 

We believe that this revised version, which includes reviewers’ suggestions, is more accurate and communicates better the main message of the article. 

We will gratefully respond to any further comments on the text.

The article has been linguistically proofread by a professional proof reader. In the manuscript, the corrected text is highlighted in yellow. 

Thank you again for taking the time to review our paper and for your constructive comments.

Best regards,

Iwona Malinowska-Lipień

---

## [Decision Letter · Decision Letter 3]

22 Nov 2021

Nurses and physicians attitudes towards factors related to hospitalized patient safety

PONE-D-21-20381R3

Dear Dr. Malinowska-Lipień,

We’re pleased to inform you that your manuscript has been judged scientifically suitable for publication and will be formally accepted for publication once it meets all outstanding technical requirements.

Kind regards,

Sandul Yasobant, PhD

Academic Editor

PLOS ONE

Additional Editor Comments (optional):

Reviewers' comments:

Reviewer's Responses to Questions

**Comments to the Author**

1. If the authors have adequately addressed your comments raised in a previous round of review and you feel that this manuscript is now acceptable for publication, you may indicate that here to bypass the “Comments to the Author” section, enter your conflict of interest statement in the “Confidential to Editor” section, and submit your "Accept" recommendation.

Reviewer #2: All comments have been addressed

2. Is the manuscript technically sound, and do the data support the conclusions?

Reviewer #2: Yes

3. Has the statistical analysis been performed appropriately and rigorously? 

Reviewer #2: Yes

4. Have the authors made all data underlying the findings in their manuscript fully available?

Reviewer #2: Yes

5. Is the manuscript presented in an intelligible fashion and written in standard English?

Reviewer #2: Yes

6. Review Comments to the Author

Reviewer #2: Almost all comments were addressed properly and i still suggest you to improve your english and grammer.

7. PLOS authors have the option to publish the peer review history of their article (what does this mean?). If published, this will include your full peer review and any attached files.

Reviewer #2: No

---

## [Editor Report · Acceptance letter]

24 Nov 2021

PONE-D-21-20381R3 

Nurses and physicians attitudes towards factors related to hospitalized patient safety 

Dear Dr. Malinowska-Lipień:

I'm pleased to inform you that your manuscript has been deemed suitable for publication in PLOS ONE. Congratulations! Your manuscript is now with our production department. 

Kind regards, 

on behalf of

Dr. Sandul Yasobant 

Academic Editor

PLOS ONE